# Detection of rs665862918 (15-bp Indel) of the *HIAT1* Gene and its Strong Genetic Effects on Growth Traits in Goats

**DOI:** 10.3390/ani10020358

**Published:** 2020-02-23

**Authors:** Jiayang Gao, Xiaoyue Song, Hui Wu, Qi Tang, Zhenyu Wei, Xinyu Wang, Xianyong Lan, Bao Zhang

**Affiliations:** 1College of Medicine & Forensic, Health Science Center, Xi’an Jiaotong University, Xi’an 710061, Shaanxi, China; jiayangzth12@163.com; 2Shaanxi Key Laboratory of Molecular Biology for Agriculture, College of Animal Science, Northwest A&F University, Yangling 712100, Shaanxi, China; wuhui582@163.com (H.W.); tangqi960@163.com (Q.T.); weizhenyu0222@163.com (Z.W.); wangxinyu6157@163.com (X.W.); 3Shaanxi Provincial Engineering and Technology Research Center of Cashmere Goats, Yulin University, Yulin 719000, Shaanxi, China; songxiaoyue@yulinu.edu.cn; 4Life Science Research Center, Yulin University, Yulin 719000, Shaanxi, China

**Keywords:** goat, *HIAT1* gene, indel, growth trait, association

## Abstract

**Simple Summary:**

Growth traits are important in goats and can affect their body size and meat production. In this study, the hippocampus abundant transcript 1 (*HIAT1*) gene, which has been reported as a meat-associated trait in elite goat breeds and a dairy-associated trait in water buffalo, was chosen to detect its correlation with growth traits in goats. The results show that the rs665862918 polymorphism (a 15 bp insertion) in *HIAT1* is associated with body length, chest width, chest depth, height at hip cross and heart girth in Shaanbei white cashmere goats (SBWC, *n* = 1013). Our results reveal that rs665862918 in *HIAT1* is relevant to the growth traits of goats and could be used for marker-assisted selection (MAS) as a molecular marker in goat populations.

**Abstract:**

The hippocampus abundant transcript 1 (*HIAT1*) gene, which was detected by the genome-wide identification of selective sweeps among elite goat breeds and water buffalo, is proposed to play an important role in meat characteristics. Four indels of the *HIAT1* gene selected from the NCBI and Ensembl databases were detected via a pooling and sequencing strategy. A 15 bp insertion (rs665862918) in the first intron of *HIAT1* was selected and classified on an electrophoresis platform in the Shaanbei white cashmere goat (SBWC) population. The correlation analysis revealed that rs665862918 is significantly highly associated with chest width (*p* = 1.57 × 10^−5^), chest depth (*p* = 8.85 × 10^−5^), heart girth (*p* = 1.05 × 10^−7^), body length (*p* = 0.022), and height at hip cross (*p* = 0.023) in the SBWC population (*n* = 1013). Further analysis revealed that individuals with a genotype insertion/insertion (II) of the rs665862918 locus exhibited better growth trait performance than individuals with an insertion/deletion (ID) or deletion/deletion (DD). These findings verify that *HIAT1* affects the body size of goats and that rs665862918 could be a potential molecular marker for growth traits in goat breeding.

## 1. Introduction

The goat was the first animal domesticated for consumer production and plays an important role in the food chain. The present goat breeding programs aim to improve the growth rate and prolificacy of goats through pure breeding and the selection of indigenous breeds for skin, meat, and milk traits [1,2]. Growth traits of goats, such as their body length, body height, and heart girth, are critical factors that determine goat production [3,4,5]. Molecular breeding, which exploits functional single nucleotide polymorphisms (SNPs), insertions/deletions (indels), and copy number variations (CNVs) of candidate genes, is a reproductive approach for improving the growth traits of goats [3,6]. In contrast to other types of variations, indels are a type of natural variation in genes that refer to one DNA chain with a certain number of nucleotide insertions or deletions in the genome [7]. Indel variants present the advantages of convenient detection and causing notable effects compared with other types of variations (such as SNPs, CNVs, and structural variations in the genome) [8,9]. Currently, indels are detected on an electrophoresis platform, which is an economical and fast method that does not require complicated experimental equipment [10,11,12].

Shaanbei white cashmere goats (SBWC), resulting from a cross between Shaanbei black goat (female parent) and Liaoning cashmere goat (male parent), are a rustic breed that exhibits increased resistance to rough feed, wind, cold weather, and disease (Figure 1) [5]. These goats are widely raised in the northern Shaanxi province of China, exhibiting excellent cashmere performance and well-established meat quality [13,14]. However, due to its short stature, SBWC shows lower meat production compared with other well-known breeds. Thus, the improvement of SBWC products by detecting important candidate gene polymorphisms and their effect on growth-related traits represents a potentially rewarding approach.

The *HIAT1* gene was first identified as an abundant hippocampus transcript and classified as a member of the major facilitator superfamily of solute carrier proteins (SLCs) [15,16]. The SLCs include a large group of proteins that transport diverse substances, including amino acids, sugars, nucleosides and fatty acids [16,17,18]. The *HIAT1* gene, which is also known as *MFSD14A* (major facilitator superfamily domain containing 14A), plays a role in transmembrane transport and the molecular function of transporter activity [18]. Our previous genome resequencing study of goats revealed that a set of genes play critical roles in meat goats compared with wool and dairy breeds, including the *HIAT1* gene [19]. In addition, *HIAT1* was identified as a novel candidate gene for milk production in buffaloes and was localized to a bovine QTL (quantitative trait locus) affecting the milk fat yield and protein yield [20,21]. *HIAT1* may transport a bloodstream solute that is required for the final stages of spermatogenesis in mice [22]. Overall, *HIAT1* has been proposed to play important roles at specific loci through transporter activity and in the modulation of mammalian growth progress, especially in skeletal development [6]. Therefore, we evaluated whether the *HIAT1* gene is associated with growth traits in goats. These findings provide potential molecular markers for marker-assisted selection (MAS) programs to improve the production of indigenous breeds in the goat industry.

## 2. Materials and Methods

The experimental procedures were approved by the Review Committee of the Health Science Center of Xi’an Jiaotong University (XJTU, project identification code 2013–054). Animal experiments and sample collection were performed following the ethics commission’s guidelines.

### 2.1. Samples and Collection of Data

In this study, 1013 uncorrelated female SBWC goats (2–3 years old) were used to obtain samples from ear tissue, which were collected from different farms maintained under similar management plans, environmental conditions, and feeding programs in central Yulin, Shaanxi, China. Body height (BH), body length (BL), chest width (CW), chest depth (CD), heart girth (HG), cannon circumference (CC), and height at hip cross (HHC) were recorded as growth traits according to the protocol of Gilbert, et al. [23].

### 2.2. Isolation of DNA and Primer Design

Genomic DNA samples isolated via a high-salt extraction method from the ear tissues of 1013 SBWC goats [24] were quantified with a NanoDrop 1000 spectrophotometer (Thermo Scientific, Waltham, MA, USA). The DNA samples were diluted to 50 ng/µL as working solutions [3].

Data on the indels present in *HIAT1* in goats were downloaded from Ensembl (http://asia.ensembl.org/index.html). The *HIAT1* gene of goat is in chromosome 3 with 46,942 bp, including 12 exons and 11 introns. All the sequences of the *HIAT1* gene were analyzed for indels, 112 indels were blasted, and 4 indels with appropriate lengths and locations were selected: rs658176778 (in the 5′ UTR, a 14 bp deletion), rs665862918 (in the intron 1, a 15 bp insertion), rs672419140 (in the intron 1, an 8 bp insertion), and rs668363704 (in the intron 2, an 8 bp insertion). Then, Primer Premier software (Version 5.0, Premier Biosoft, San Francisco, CA, USA) was used to design four primer pairs according to the reference sequence (GenBank NC_030810, Table 1).

### 2.3. DNA Pool Establishment and PCR Amplification

A genomic DNA pool was constructed from 30 randomly selected DNA samples from SBWC goats, consisting of a mixture of 1 µL of each sample, and then amplification was performed with 4 pairs of primers. The polymerase chain reaction (PCR) products were sequenced and analyzed according to Wang’s method for detecting polymorphisms [4].

PCR was performed in a total volume of 13 µL, containing 0.6 µL (50 ng/µL) of genomic DNA, 5.1 µL of ddH_2_O, 0.4 µL of each primer, and 6.5 µL of 2× Taq Master Mix. Touch-down PCR (TD-PCR) was applied in the thermal cycling program as described in Yang’s study [25]. Indel polymorphisms were assayed by subjecting 5.0 µL of the PCR products to electrophoresis on 3.0% agarose gels [5].

### 2.4. Statistical Analysis

The genotypic and allelic frequencies of indel variants were calculated directly by hand. The Hardy–Weinberg equilibrium (HWE) was calculated with the SHEsis program (http://analysis.bio-x.cn/myAnalysis.php). Population genetic parameters including homozygosity (Ho), heterozygosity (He), polymorphism information content (PIC), and the effective allele number (Ne) were computed using Popgene software (version 1.3.1, University of Alberta, Edmonton, AB, Canada) [26]. The associations between the indel polymorphisms and growth traits were evaluated by one-way analysis of variance (ANOVA) with SPSS software (version 21.0, IBM, Armonk, NY, USA) [27]. The results are presented in format of means ± standard errors.

## 3. Results

### 3.1. Electrophoresis Detection of Four Indels

According to the results of the DNA pool amplification, electrophoresis, and sequencing, two indels of *HIAT1* were found to be polymorphic: the 15 bp insertion (NC_030810.1:g.7695 insACTAGTGGACTTCTT, rs665862918) in intron 1 and the 8 bp insertion (NC_030810.1:g.20531 ins ATCGGGTT, rs668363704) in intron 2. The rs665862918 locus was chosen because the 15 bp insertion was located in intron 1, closer to the H3K27ac mark which was found to distinguish active enhancers from inactive/poised enhancers based on the UCSC genome database (http://genome.ucsc.edu/) and therefore more likely to be functional. DNA sequencing and alignment based on the reference sequence (NC_030810.1) were used to identify the indel (Figure 2a,b).

As shown in Figure 2c, rs665862918 presented three genotypes according to agarose gel electrophoresis: homozygote deletion type (deletion/deletion: DD, 183 bp), insertion type (insertion/insertion: II, 198 bp), and heterozygote type (insertion/deletion: ID, 198 bp and 183 bp). The band accompanying the heterozygote type is referred to as a heteroduplex, which is a double-stranded DNA molecule formed by the base pairing of complementary single strands of different double-stranded parent molecules and can be produced via intracellular genetic recombination or form under in vitro renaturation conditions [28]. Heteroduplexes were reported in a previous study in which indel polymorphisms were detected by agarose gel electrophoresis [29].

### 3.2. Polymorphism of rs665862918

The II, ID, and DD genotypes of rs665862918 were detected in SBWC, and the frequency of DD was higher than those of ID and II in 1013 female individuals. The frequencies of the I and D alleles were 0.203 and 0.797, respectively. No significant deviations from Hardy–Weinberg equilibrium were found in the cases or controls (*p* = 0.520, *p* > 0.05). Intermediate genetic diversity was calculated according to the PIC value (PIC = 0.271, 0.25 <PIC< 0.5) at this locus (Table 2).

### 3.3. Effect of the rs665862918 Polymorphism on Prowth Traits

rs665862918 was found to be significantly related to chest width, chest depth, heart girth, body length, and the height at hip cross (*p* = 0.022, *p* = 1.57 × 10^−5^, *p* = 8.85 × 10^−5^, *p* = 1.05 × 10^−7^, and *p* = 0.023, respectively) in SBWC goats. The body length of genotype II goats was greater than that of genotype DD goats (*p* < 0.05). The chest width associated with genotype II was greater than that associated with genotypes ID (*p* < 0.05) and DD (*p* < 0.01). The chest depth associated with genotype II was greater than those of genotypes ID (*p* < 0.05) and DD (*p* < 0.01). The heart girth of genotype II goats was greater than those of genotype ID (*p* < 0.05) and DD (*p* < 0.01) goats. The height at hip cross associated with genotype II was greater than that associated with genotype DD (*p* < 0.05) (Table 3).

## 4. Discussion

In previous studies, SNPs in genes were shown to be associated with growth traits and litter size in goats [30,31,32,33]. Indels similarly present relationships with growth traits [34,35], as observed for a 12 bp indel in *GDF9* that is related to first-born litter size in goats [4], and a 14 bp indel in *CMTM2* that shows a relationship with litter size in goats [24]. In this study, rs665862918 of the *HIAT1* gene was found to be associated with the growth traits of SBWC goats. *HIAT1* is located on chromosome 3 including 12 exons and 11 introns, and the indel position ranges from 77,286,424 to 77,286,439 bp. Here, the rs665862918 (15 bp indel) of *HIAT1* in SBWC goats was significantly correlated with body length (*p* = 0.022), chest width (*p* = 1.57 × 10^−5^), chest depth (*p* = 8.85 × 10^−5^), heart girth (*p* = 1.05 × 10^−7^), and height at hip cross (*p* = 0.023).

There are several reasons for addressing these associations. First, a region on chromosome 3 was reported to show the highest intensity signal effect on meat traits in goats, which includes the *HIAT1*, *SLC35A3*, and *SASS6* genes [19,21]. The same region spanning 43.3 to 43.8 Mb on chromosome 3 of water buffalo was shown to affect milk fat and protein percentages, and this region harbors the *HIAT1*, *SLC35A3*, and *PALMD* genes [21]. A region on chromosome 3 at 43.29 Mb was also identified as a bovine QTL affecting milk fat yield, fat percentage, protein yield, and protein percentage [20]. Whitworth, et al. [36] found that *HIAT1* downregulates transcripts related to nuclear transfer in the pig blastocyst stage and reported that the gene might participate in mouse spermatogenesis [22]. However, functional studies of *HIAT1* are limited. We showed that rs665862918 of *HIAT1* within SBWC goats is highly significantly correlated with growth traits, which implied that *HIAT1* may be a novel candidate gene that has genetic effects on meat production in goats. Second, the neighboring gene *SLC35A3* was reported to be associated with vertebral malformation [37] and was also reported to affect milk production and milk performance in dairy cattle [38,39,40]. In addition, the *HIAT1* and *SLC35A3* genes in goats, cattle, and water buffalo are all highly selected, which suggests that *HIAT1* and *SLC35A3* may work together to affect certain characteristics. Third, the alternative splicing of eukaryotic transcripts causes cells to generate highly diverse proteins from a limited number of genes by generating different mRNAs [41]. According to the Ensembl database, eight transcripts have been detected in *HIAT1* of humans, whereas only one transcript has been reported in goats. Since transcripts show high consistency in different species, the question arises as to whether rs665862918 in intron 1 of the *HIAT1* gene may affect gene expression through different transcripts.

There are several limitations to interpreting our results. As a dual-purpose variety, SBWC is used to produce both cashmere and meat. Due to the lack of data from cashmere goats, only the relationship between growth traits and the indels of *HIAT1* was studied. In addition, 1013 samples from SBWC were tested in the study; if the results are to be persuasive, more samples from male goats and other breeds will be indispensable.

## 5. Conclusions

In this study, the rs665862918 polymorphism in *HIAT1* was significantly associated with body length, the height at hip cross, chest depth, chest width, and heart girth in a large SBWC population. The results presented in this paper suggested that rs665862918 could be a potential marker for MAS, and further studies and replication with different goat breeds will be required to improve the growth traits of goats.

## Figures and Tables

**Figure 1 animals-10-00358-f001:**
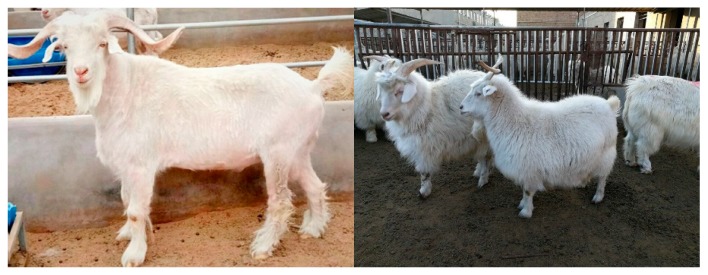
Shaanbei white cashmere goats.

**Figure 2 animals-10-00358-f002:**
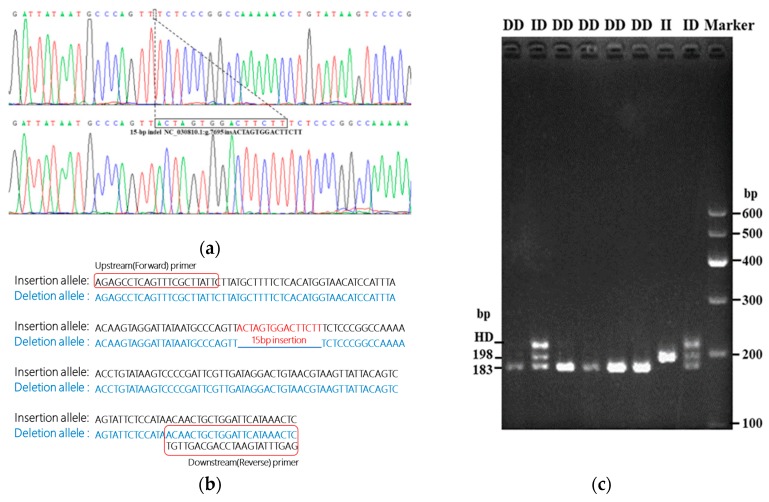
(**a**) Sequence diagram of the rs665862918 mutations of the goat *HIAT1* gene. (**b**) Analysis of the 15 bp indel sequence of the goat *HIAT1* gene in reference to NC_030810.1. (**c**) Electrophoresis assay of the rs665862918 locus of the *HIAT1* gene of goat (3% agarose gel electrophoresis). HD, heteroduplex.

**Table 1 animals-10-00358-t001:** PCR primers used for detecting indels in the goat *HIAT1* gene.

Primers	Location	Primer Sequences (5′ to 3′)	Length (bp)	Tm
rs658176778	5′ UTR	F:ATAGCATGGACAGAGGAGCCT	192/178	TD-PCR
		R:TCCCTGGTAAAGAACAGCAAG		
rs665862918	Intron	F:AGAGCCTCAGTTTCGCTTATT	183/198	TD-PCR
		R:GAGTTTATGAATCCAGCAGTTGT		
rs672419140	Intron	F:GGATGACAGAGGATGAGATGG	137/145	TD-PCR
		R:CAGTCGTGTCTGACTCTTTGTG		
rs668363704	Intron	F:GTTAGGCAGCAATAGCTCAAGG	166/158	TD-PCR
		R:AAAACCCAACAAATGGAAGATG		

TD-PCR: Touch-down PCR; Tm: melting temperature.

**Table 2 animals-10-00358-t002:** PCR primers used for detecting indels in the goat *HIAT1* gene.

Genotype Frequencies		Allele Frequencies		HWE	Population Parameters			
				*p*-Value	Ho	He	Ne	PIC
DD	0.639 (*n* = 647)	I	0.203	0.520	0.677	0.323	1.478	0.271
ID	0.317 (*n* = 321)							
		D	0.797					
II	0.044 (*n* = 45)							

Hardy–Weinberg equilibrium (HWE), homozygosity (Ho), heterozygosity (He), effective allele number (Ne), and polymorphism information content (PIC).

**Table 3 animals-10-00358-t003:** The relationship between the *HIAT1* gene and growth traits in Shaanbei white cashmere goats (*p* < 0.05/*p* < 0.01).

Traits	Observed Genotypes (Least Squares Means)			*p*-Values
	DD (647)	ID (321)	II (45)	
BH (cm)	56.22 ± 0.18	56.48 ± 0.28	57.76 ± 0.78	0.092
**BL (cm)**	63.76 ^b^ ± 0.20	64.29 ^a,b^ ± 0.32	66.73 ^a^ ± 1.04	0.022
**CW (cm)**	18.87 ^B^ ± 0.14	19.85 ^A^ ± 0.24	20.92 ^A^ ± 0.37	1.57 × 10^−5^
**CD (cm)**	26.80 ^B^ ± 0.14	27.48 ^A^ ± 0.21	28.94 ^A^ ± 0.63	8.85 × 10^−5^
**HG (cm)**	79.79 ^B^ ± 0.44	83.53 ^A^ ± 0.66	86.56 ^A^ ± 1.81	1.05 × 10^−7^
**HHC (cm)**	58.58 ^b^ ± 0.21	59.18 ^a,b^ ± 0.35	60.81 ^a^ ± 0.87	0.023
CC (cm)	7.85 ± 0.11	7.96 ± 0.05	8.10 ± 0.11	0.660

BH, body height; BL, body length; CW, chest width; CD, chest depth; HG, heart girth; HHC, height at hip cross; CC, cannon circumference [4]. (a, b, *p* < 0.05; A, B, *p* < 0.01). The rows in bold represent parameters with significant difference between different genotypes.

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
