# Peer review of "Detection of rs665862918 (15-bp Indel) of the HIAT1 Gene and its Strong Genetic Effects on Growth Traits in Goats"

_animals, 2020, doi:10.3390/ani10020358_

Round 1
Reviewer 1 Report
Line 94-95: please describe shortly how indel selection have been addressed. These mutations are all the indels found in the gene? All the gene sequence has been analysed or only a particular range? If these are only few of the indels that could exist in this gene justify the choice.
Line 95: how “with appropriate lengths and locations” is defined? Appropriate for what reason?
Line 117-120: an ANOVA analysis should be adequate for the purpose of the present study but some detail should be reported and some preliminary check should be performed before ANOVA application. For example, number of animals for each group (genotype) is very different due the different frequency and this could be a problem for simple ANOVA besides Normality and other similar prerequisite should be tested before in order to choose parametric or not parametric analysis. Age of animals have been considered in the analysis? In this kind of variables could be significative because an animal of 2 year is in a different philological status than 3 years old one.
Line 127 pag 4: please report some numerical indication of the nearest position of the indel to the gene.
Line 129 pag 4: why the chosen indel was easier to be genotyped? According to the table 1 the difference in size of each indel is 14,15,8 and 8 respectively that are normally a sufficient size to be see in a common agarose gel without any particular modification. If other technical problem has been used to justify this choice should be reported.
Author Response
Line 94-95: please describe shortly how indel selection have been addressed. These mutations are all the indels found in the gene? All the gene sequence has been analysed or only a particular range? If these are only few of the indels that could exist in this gene justify the choice.
Line 95: how “with appropriate lengths and locations” is defined? Appropriate for what reason?
Response: Thanks for your comments and advice! These two questions above will be answered together. Firstly, these four indels are all from the gene HIAT1, and were selected from the database of ensembl (http://asia.ensembl.org/index.html).
Second, the gene HIAT1 of goat is in chrome 3 with 46,942 bp, included 12 exons and 11 introns. All the sequence of HIAT1 gene had been analyzed for indels,112 indels are blasted.
The principles of indels are selected as following criteria: ① Indel should showed polymorphism by DNA pool detection in electrophoresis platform;② Indel are close to promoter, terminator or enhancer, which means the indels more likely to be functional in gene expression;③the length of indels are larger than 8 bp, which is easier to detect in gel electrophoresis.
Among the 112 indels, 7 indels are more than 8 bp. We designed 7 pair of primers, only 4 of them can obtain amplicons and detected in the electrophoresis platform.
Finally, rs665862918 loci was chosen by two reasons: first, the rs665862918 insertion was closer to the H3K27ac mark which was found to distinguish active enhancers from inactive/poised enhancers based on the genome database of UCSC (http://genome.ucsc.edu/); second, 15-bp fragment of rs665862918 loci was more easier to detect by the electrophoresis platform.
Line 117-120: an ANOVA analysis should be adequate for the purpose of the present study but some detail should be reported and some preliminary check should be performed before ANOVA application. For example, number of animals for each group (genotype) is very different due the different frequency and this could be a problem for simple ANOVA besides Normality and other similar prerequisite should be tested before in order to choose parametric or not parametric analysis. Age of animals have been considered in the analysis? In this kind of variables could be significative because an animal of 2 year is in a different philological status than 3 years old one.
Response: Thank you for asking me these questions. Firstly, the individual number of genotype II is lower than DD and ID, because the frequency of II is lower in this locus. But it’s adequate to use ANOVA analysis for the purpose of present study, which is still fit the protocol of statistics. Although the individual number of II is 45, it’s enough for ANOVA statistics with little significant errors.
Second, age of goats in this experiment is between 2 and 3 years old, they both belong to adult goats and can be put in the same study group. There are not significant differences of growth traits between goats with 2 and 3 years old. Previews studies used this group to analysis in goats, some of them are listed below.
Wang, K.; Yan, H.; Xu, H.; Yang, Q.; Zhang, S.; Pan, C.; Chen, H.; Zhu, H.; Liu, J.; Qu, L., et al. A novel indel within goat casein alpha S1 gene is significantly associated with litter size. Gene 2018, 671, 161-169, doi:10.1016/j.gene.2018.05.119. Wang, X.; Yang, Q.; Wang, K.; Zhang, S.; Pan, C.; Chen, H.; Qu, L.; Yan, H.; Lan, X. A novel 12-bp indel polymorphism within the GDF9 gene is significantly associated with litter size and growth traits in goats. Animal genetics 2017, 48, 735-736, doi:10.1111/age.12617. Wang, Z.; Zhang, X.; Jiang, E.; Yan, H.; Zhu, H.; Chen, H.; Liu, J.; Qu, L.; Pan, C.; Lan, X. InDels within caprine IGF2BP1 intron 2 and the 3’-untranslated regions are associated with goat growth traits. Animal genetics 2019, 10.1111/age.12871, doi:10.1111/age.12871.
Line 127 pag 4: please report some numerical indication of the nearest position of the indel to the gene.
Response: Thanks for your comments.rs665862918 is located in 7,695 bp, intron 1 of goat HIAT1(NC_030810.1:g.7695 insACTAGTGGACTTCTT, rs665862918). And the location of H3K27acmarker is located in promoter and intron 1 region in the human HIAT1 according to UCSC database.
Line 129 pag 4: why the chosen indel was easier to be genotyped? According to the table 1 the difference in size of each indel is 14,15,8 and 8 respectively that are normally a sufficient size to be see in a common agarose gel without any particular modification. If other technical problem has been used to justify this choice should be reported.
Response: Thank you for your comments, in this study, 4 indels was selected and detected, only 2 of them are polymorphism: rs665862918, a 15-bp insertion in intron1 and rs668363704, an 8-bp insertion in intron 2. The 15-bp insertion was chosen because the indel in intron 1 is more likely functional. There isn’t other technical problem has been used to justify this choice.
Reviewer 2 Report
My evaluation report of this manuscript was delayed because I did not find the reviewer response note in the journal system, which was later submitted to me by the journal staff.
The manuscript from Gao and others was substantially improved in this revised version. Considering the initial evaluation and all the changes and take into consideration the limitations of the work, I recommend the publication as a short communication after minor revision and the inclusion of relevant work in the discussion.
There is two newly-published paper that must be included in the discussion of this manuscript since it deals with the same theme:
https://www.mdpi.com/2076-2615/10/1/168
https://www.mdpi.com/2076-2615/9/12/1114
Additionally, these recent papers could be cited in the introduction or discussion since the manuscript has several citations from works with different species:
https://www.mdpi.com/2076-2615/10/1/75
https://www.mdpi.com/2076-2615/9/11/910
Minor but mandatory corrections
The figures are wrong in the revised text. Please add a text in the methodology explaining what is figure 1 before the figure shown in the paper and correct the text because you are naming the figure 2 as figure 1 (lines 131 and 132).
At all tables, please define all abbreviations used in the table at the footnote of each table. Also, inform the statistical test for the p-valued presented in the table.
Check your references, the journal name it is "Animals" not "Animals: an open-access journal from MDPI"
Author Response
My evaluation report of this manuscript was delayed because I did not find the reviewer response note in the journal system, which was later submitted to me by the journal staff.
The manuscript from Gao and others was substantially improved in this revised version. Considering the initial evaluation and all the changes and take into consideration the limitations of the work, I recommend the publication as a short communication after minor revision and the inclusion of relevant work in the discussion.
There is two newly-published paper that must be included in the discussion of this manuscript since it deals with the same theme:
https://www.mdpi.com/2076-2615/10/1/168
https://www.mdpi.com/2076-2615/9/12/1114
Additionally, these recent papers could be cited in the introduction or discussion since the manuscript has several citations from works with different species:
https://www.mdpi.com/2076-2615/10/1/75
https://www.mdpi.com/2076-2615/9/11/910
Response: Thank you for your comments and advice, and these 4 papers mentioned above have already cited in the manuscript.
Minor but mandatory corrections
The figures are wrong in the revised text. Please add a text in the methodology explaining what is figure 1 before the figure shown in the paper and correct the text because you are naming the figure 2 as figure 1 (lines 131 and 132).
Response: Thank you for your comments, and we have reframed the article yet follow your advice.
At all tables, please define all abbreviations used in the table at the footnote of each table. Also, inform the statistical test for the p-valued presented in the table.
Response: Thank you for your comments, and we have reframed the article yet follow your advice.
Check your references, the journal name it is "Animals" not "Animals: an open-access journal from MDPI"
Response: Thank you for your comments, and we have reframed the article yet follow your advice.
Round 2
Reviewer 1 Report
Dear author,
thank you for reviewing the article. Please find few new observation at the end of each response.
Line 94-95: please describe shortly how indel selection have been addressed. These mutations are all the indels found in the gene? All the gene sequence has been analysed or only a particular range? If these are only few of the indels that could exist in this gene justify the choice.
Line 95: how “with appropriate lengths and locations” is defined? Appropriate for what reason?
Response: Thanks for your comments and advice! These two questions above will be answered together. Firstly, these four indels are all from the gene HIAT1, and were selected from the database of ensembl (http://asia.ensembl.org/index.html).
Second, the gene HIAT1 of goat is in chrome 3 with 46,942 bp, included 12 exons and 11 introns. All the sequence of HIAT1 gene had been analysed for indels,112 indels are blasted.
The principles of indels are selected as following criteria: ① Indel should showed polymorphism by DNA pool detection in electrophoresis platform;② Indel are close to promoter, terminator or enhancer, which means the indels more likely to be functional in gene expression;③the length of indels are larger than 8 bp, which is easier to detect in gel electrophoresis.
Among the 112 indels, 7 indels are more than 8 bp. We designed 7 pair of primers, only 4 of them can obtain amplicons and detected in the electrophoresis platform.
Finally, rs665862918 loci was chosen by two reasons: first, the rs665862918 insertion was closer to the H3K27ac mark which was found to distinguish active enhancers from inactive/poised enhancers based on the genome database of UCSC (http://genome.ucsc.edu/); second, 15-bp fragment of rs665862918 loci was more easier to detect by the electrophoresis platform.
Thank you for your response. Please add this sentence to the text “Second, the gene HIAT1 of goat is in chrome 3 with 46,942 bp, included 12 exons and 11 introns. All the sequence of HIAT1 gene had been analysed for indels,112 indels are blasted.” And/or better report as supplementary material all the indel retrieved by blast analysis.
“
Line 117-120: an ANOVA analysis should be adequate for the purpose of the present study but some detail should be reported and some preliminary check should be performed before ANOVA application. For example, number of animals for each group (genotype) is very different due the different frequency and this could be a problem for simple ANOVA besides Normality and other similar prerequisite should be tested before in order to choose parametric or not parametric analysis. Age of animals have been considered in the analysis? In this kind of variables could be significative because an animal of 2 year is in a different philological status than 3 years old one.
Response: Thank you for asking me these questions. Firstly, the individual number of genotype II is lower than DD and ID, because the frequency of II is lower in this locus. But it’s adequate to use ANOVA analysis for the purpose of present study, which is still fit the protocol of statistics. Although the individual number of II is 45, it’s enough for ANOVA statistics with little significant errors.
Second, age of goats in this experiment is between 2 and 3 years old, they both belong to adult goats and can be put in the same study group. There are not significant differences of growth traits between goats with 2 and 3 years old. Previews studies used this group to analysis in goats, some of them are listed below.
Thank you for your response. In my opinion statistical analysis can be still improved. Here some point that should in my opinion be considerate. 1) if age of animals has no effect it should be demonstrated, if it is the same population used in other study it should be reported, in particular I would include age, farm as factor. Also the effect of allele substitution using the Falconer procedure can be applied (here one recent article of these issue https://www.ncbi.nlm.nih.gov/pubmed/25023801).
Line 127 pag 4: please report some numerical indication of the nearest position of the indel to the gene.
Response: Thanks for your comments.rs665862918 is located in 7,695 bp, intron 1 of goat HIAT1(NC_030810.1:g.7695 insACTAGTGGACTTCTT, rs665862918). And the location of H3K27acmarker is located in promoter and intron 1 region in the human HIAT1 according to UCSC database.
Thank you for your comments, please report in the text the information.
Line 129 pag 4: why the chosen indel was easier to be genotyped? According to the table 1 the difference in size of each indel is 14,15,8 and 8 respectively that are normally a sufficient size to be see in a common agarose gel without any particular modification. If other technical problem has been used to justify this choice should be reported.
Thank you for your comments, in this study, 4 indels was selected and detected, only 2 of them are polymorphism: rs665862918, a 15-bp insertion in intron1 and rs668363704, an 8-bp insertion in intron 2. The 15-bp insertion was chosen because the indel in intron 1 is more likely functional. There isn’t other technical problem has been used to justify this choice.
Thank you for your response, so please change in the text “rs665862918 locus was easier to detect” with “4 indels was selected and detected, only 2 of them are polymorphism: rs665862918, a 15-bp insertion in intron1 and rs668363704, an 8-bp insertion in intron 2. The 15-bp insertion was chosen because the indel in intron 1 is more likely functional” or something similar
Author Response
Dear reviewer,
Thanks for your comments and advice! The manuscript is modified as follows:
Line 94-95: please describe shortly how indel selection have been addressed. These mutations are all the indels found in the gene? All the gene sequence has been analysed or only a particular range? If these are only few of the indels that could exist in this gene justify the choice.
Line 95: how “with appropriate lengths and locations” is defined? Appropriate for what reason?
Response: Thanks for your comments and advice! These two questions above will be answered together. Firstly, these four indels are all from the gene HIAT1, and were selected from the database of ensembl (http://asia.ensembl.org/index.html).
Second, the gene HIAT1 of goat is in chrome 3 with 46,942 bp, included 12 exons and 11 introns. All the sequence of HIAT1 gene had been analysed for indels,112 indels are blasted.
The principles of indels are selected as following criteria: ① Indel should showed polymorphism by DNA pool detection in electrophoresis platform;② Indel are close to promoter, terminator or enhancer, which means the indels more likely to be functional in gene expression;③the length of indels are larger than 8 bp, which is easier to detect in gel electrophoresis.
Among the 112 indels, 7 indels are more than 8 bp. We designed 7 pair of primers, only 4 of them can obtain amplicons and detected in the electrophoresis platform.
Finally, rs665862918 loci was chosen by two reasons: first, the rs665862918 insertion was closer to the H3K27ac mark which was found to distinguish active enhancers from inactive/poised enhancers based on the genome database of UCSC (http://genome.ucsc.edu/); second, 15-bp fragment of rs665862918 loci was more easier to detect by the electrophoresis platform.
Thank you for your response. Please add this sentence to the text “Second, the gene HIAT1 of goat is in chrome 3 with 46,942 bp, included 12 exons and 11 introns. All the sequence of HIAT1 gene had been analysed for indels,112 indels are blasted.” And/or better report as supplementary material all the indel retrieved by blast analysis.
Response: thanks for your comments, and the manuscript was reframed with your advice. (Line 95-97)
Line 117-120: an ANOVA analysis should be adequate for the purpose of the present study but some detail should be reported and some preliminary check should be performed before ANOVA application. For example, number of animals for each group (genotype) is very different due the different frequency and this could be a problem for simple ANOVA besides Normality and other similar prerequisite should be tested before in order to choose parametric or not parametric analysis. Age of animals have been considered in the analysis? In this kind of variables could be significative because an animal of 2 year is in a different philological status than 3 years old one.
Response: Thank you for asking me these questions. Firstly, the individual number of genotype II is lower than DD and ID, because the frequency of II is lower in this locus. But it’s adequate to use ANOVA analysis for the purpose of present study, which is still fit the protocol of statistics. Although the individual number of II is 45, it’s enough for ANOVA statistics with little significant errors.
Second, age of goats in this experiment is between 2 and 3 years old, they both belong to adult goats and can be put in the same study group. There are not significant differences of growth traits between goats with 2 and 3 years old. Previews studies used this group to analysis in goats, some of them are listed below.
Thank you for your response. In my opinion statistical analysis can be still improved. Here some point that should in my opinion be considerate. 1) if age of animals has no effect it should be demonstrated, if it is the same population used in other study it should be reported, in particular I would include age, farm as factor. Also the effect of allele substitution using the Falconer procedure can be applied (here one recent article of these issue https://www.ncbi.nlm.nih.gov/pubmed/25023801).
Response: thanks for your comments. Firstly, age of goats may have great effects before 1.5 years old, but when goats in their adulthood (after 1.5 years old), the effects in growth traits are not significant anymore, and goats with 2-3 years old were used in many studies and the results are stable and reliable [1-2]. Besides, the 1013 Shaanbei white cashmere goats are raised in the same farm and with same conditions, so that to minimize the error.
- Tang, Q.; Zhang, X.L.; Wang, X.Y.; Wang, K.; Yan, H.L.; Zhu, H.J.; Lan, X.Y.; Lei, Q.; Pan, C.Y. Detection of two insertion/deletions (indels) within the ADAMTS9 gene and their associations with growth traits in goat. Small Ruminant Res. 2019, 180, 9-14, doi:10.1016/j.smallnunres.2019.09.015.
- Yan, H.L.; Jiang, E.H.; Zhu, H.J.; Hu, L.Y.; Liu, J.W.; Qu, L. The novel 22 bp insertion mutation in a promoter region of the PITX2 gene is associated with litter size and growth traits in goats. Arch Anim Breed 2018, 61, 329-336, doi:10.5194/aab-61-329-2018.
Secondly, thanks for providing the procedure, we read the paper “Influence of single nucleotide polymorphisms in the myostatin and myogenic factor 5 muscle growth-related genes on the performance traits of Marchigiana beef cattle”, thanks for your recommendation!
Line 127 pag 4: please report some numerical indication of the nearest position of the indel to the gene.
Response: Thanks for your comments.rs665862918 is located in 7,695 bp, intron 1 of goat HIAT1(NC_030810.1:g.7695 insACTAGTGGACTTCTT, rs665862918). And the location of H3K27acmarker is located in promoter and intron 1 region in the human HIAT1 according to UCSC database.
Thank you for your comments, please report in the text the information.
Response: thank you for your advice, and the information is reported in the text following your advice. (line 128-129)
Line 129 pag 4: why the chosen indel was easier to be genotyped? According to the table 1 the difference in size of each indel is 14,15,8 and 8 respectively that are normally a sufficient size to be see in a common agarose gel without any particular modification. If other technical problem has been used to justify this choice should be reported.
Thank you for your comments, in this study, 4 indels was selected and detected, only 2 of them are polymorphism: rs665862918, a 15-bp insertion in intron1 and rs668363704, an 8-bp insertion in intron 2. The 15-bp insertion was chosen because the indel in intron 1 is more likely functional. There isn’t other technical problem has been used to justify this choice.
Thank you for your response, so please change in the text “rs665862918 locus was easier to detect” with “4 indels was selected and detected, only 2 of them are polymorphism: rs665862918, a 15-bp insertion in intron1 and rs668363704, an 8-bp insertion in intron 2. The 15-bp insertion was chosen because the indel in intron 1 is more likely functional” or something similar
Response: thanks for your response, and the manuscript is reframed as your advice. (line 131- 132)